# Progress in the Medicinal Value, Bioactive Compounds, and Pharmacological Activities of *Gynostemma pentaphyllum*

**DOI:** 10.3390/molecules26206249

**Published:** 2021-10-15

**Authors:** Chao Su, Nan Li, Ruru Ren, Yingli Wang, Xiaojuan Su, Fangfang Lu, Rong Zong, Lingling Yang, Xueqin Ma

**Affiliations:** Department of Pharmaceutical Analysis, School of Pharmacy, Key Laboratory of Hui Ethnic Medicine Modernization, Ministry of Education, Ningxia Medical University, 1160 Shenli Street, Yinchuan 750004, China; scwsry@163.com (C.S.); linan950618@163.com (N.L.); luckyrenrr@163.com (R.R.); 18737803095@163.com (Y.W.); suxiaojuan96@163.com (X.S.); lffang55@163.com (F.L.); zong_rong@126.com (R.Z.)

**Keywords:** *Gynostemma pentaphyllum*, gypenosides, anti-cancer activity, anti-atherogenic effect, neuroprotective property

## Abstract

*Gynostemma pentaphyllum* (Thunb.) Makino (GP), also named Jiaogulan in Chinese, was known to people for its function in both health care and disease treatment. Initially and traditionally, GP was a kind of tea consumed by people for its pleasant taste and weight loss efficacy. With the passing of the centuries, GP became well known as more than just a tea. Until now, numbers of bioactive compounds, including saponins (also named gypenosides, GPS), polysaccharides (GPP), flavonoids, and phytosterols were isolated and identified in GP, which implied the great medicinal worth of this unusual tea. Both in vivo and in vitro tests, ranging from different cell lines to animals, indicated that GP possessed various biological activities including anti-cancer, anti-atherogenic, anti-dementia, and anti-Parkinson’s diseases, and it also had lipid-regulating effects as well as neuroprotection, hepatoprotective, and hypoglycemic properties. With the further development and utilization of GP, the research on the chemical constituents and pharmacological properties of GP were deepening day by day and had made great progress. In this review, the recent research progress in the bioactive compounds, especially gypenosides, and the pharmacological activities of GP were summarized, which will be quite useful for practical applications of GP in the treatment of human diseases.

## 1. Introduction

*Gynostemma pentaphyllum* (Thunb.) Makino is a creeping perennial herb that belongs to the family Cucurbitaceae. As a well-known edible and medicinal plant, GP has a long history of application in oriental medicine since the Ming dynasty [1]. The famous classical book of Chinese material medica, *Compendium of Materia Medica*, first recorded the usage and curative effect of GP [2]. In 1986, GP was listed by the Ministry of Science and Technology as the first “precious Chinese medicine” to be developed in the “Spark Program”. Due to its extensive biological activities, GP was brought into the list of functional foods by Ministry of Public Health of China on 5 March 2002. Currently, products containing GP had been marketed in many Asian countries and US [3]. In recent years, GP has attracted great attention owing to its wide biological activities with minimal pharmacological toxicity [4]. GP had been used in treating various diseases, such as hyperlipidemia, hyperglycemia, cancer, and so on, as shown in Figure 1. To conduct a comprehensive review, a great deal of literature on GP was obtained by searching the PubMed and China National Knowledge Infrastructure (CNKI) databases. In this paper, the research progress on chemical constituents and pharmacological properties of GP were introduced, which could provide reference for the further development of GP.

## 2. Chemical Constituents

Phytochemical studies revealed that the chemical constituents in GP included GPS, GPP, flavonoids, phytosterols, amino acids and inorganic elements, and so on (Figure 1). The detailed information of the main constituents of GP follows.

### 2.1. Gypenosides

Gypenosides isolated from GP were believed to be the major active constituents responsible for its various biological activities and reported clinical effects. In 1976, the dammarane-type saponins from GP were isolated by Japanese scholars for the first time [5]. By 2017, nearly 201 dammarane-type gypenosides had been reported in GP [6]. Since 2017, 47 new dammarane-type gypenosides had been isolated and identified, the structures are shown in Figure 2, eight of which were consistent with ginsenosides Rb1, Rb3, F2, Rg3, Rc, Rd, malonyl-Rb1, and malonyl-Rd, respectively. These ginsenosides made up around 25% of the total gypenosides in the plant, and GP was the first plant containing ginseng saponins outside of the Araliaceae family. Therefore, GP has been praised as “southern ginseng” [2]. In addition, two cucurbitane-type saponins were found in GP [7,8], and their structures are shown in Figure 3.

### 2.2. Polysaccharides, Flavonoids, Sterols, Amino Acids, and Inorganic Elements

So far, polysaccharides with various monosaccharide constituents and chemical structures have been isolated from GP. The molecular weight, monosaccharide composition, and chemical structures could be influenced by both different extraction/purification techniques [9]. Polysaccharides were mainly found in the stems and leaves of GP [10]. Ma et al. [11] determined the content of polysaccharide in different parts of GP, and the results showed that the content of polysaccharide in leaves (1.78 ± 0.60) % was higher than in stems (0.84 ± 0.23) %. GPP exhibited almost no toxic side effects on the human body but possessed broad application prospects in preventing and treating various diseases. Although flavonoids were secondary metabolites in the medicinal herb kingdom, there are few reports about flavonoids of GP. So far, more than 10 kinds of flavonoids have been reported. Ling et al. [12] investigated the comprehensive chemical profiles of GP for the first time using HPLC-ESI-QTOF-MS, and seven flavonoid glycosides were identified. From 1986 to 2020, about eighteen sterols were isolated from GP. Studies had shown that GP contained 18 kinds of amino acids, including eight kinds of essential amino acids [13]. However, the content of amino acids in GP varied from place to place. In addition, GP contained 23 kinds of inorganic elements, among which 13 kinds of trace elements and five kinds of major elements were essential to human body [13].

## 3. Pharmacological Properties of GP

### 3.1. Anti-Cancer Effect

To date, numerous studies have demonstrated the strong anti-cancer effect of the main components of GP, such as GPS, GPP, and flavonoids (as shown in Table 1). There are several mechanisms of GP’s anti-cancer effect, as shown in Figure 4.

#### 3.1.1. Immune Modulation

Tumors are closely related to the state of the body’s immune system, and strengthening immune function played an important role in the treatment of tumors. GPP (50 and 100 mg/kg for 12 days, i.g.) had an inhibitory effect on tumor growth in mouse forestomach carcinoma (MFC) gastric tumor-bearing mice, for which the tumor quality of the GPP group was significantly reduced. Furthermore, GPP could effectively prevent the immune organ atrophy and hypofunction of MFC gastric tumor-bearing mice, suggesting that the anti-tumor effect of GPP was closely associated with immune enhancement [38]. Liu et al. investigated the effect of GPS on the tumor growth in tumor-bearing mice, and the results showed that the growth of the tumor was inhibited by the intraperitoneal injection of 10–40 mg/kg GPS, and 20 mg/kg GPS exerted the strongest inhibitory effect. In addition, the number of splenic lymphocytes was increased, and the activity of natural killer (NK) cells was enhanced [39]. Studies had demonstrated that soluble Tim-3 played an important role in inhibiting the immune escape in a variety of cancer. Gypenoside XLIX (20 and 80 µg) could reduce the immune escape of non-small cell lung cancer A549 cells by promoting the stim-3/Tim-3 ratio [40].

#### 3.1.2. Induction of Cell Apoptosis

Inducing apoptosis was one of the effective ways to treat tumor, which could improve the recovery of tumor patients. Yan et al. [30] evaluated the cytotoxicity of GPS in human colorectal cancer SW-480 cells, and the results showed that GPS (70–130 µg/mL) could decrease the percentage of viable cells and increase the plasma membrane permeability of SW-480 cells in a dose- and time-dependent manner. More importantly, reactive oxygen species (ROS) play an important role in GPS-induced cell toxicity and apoptosis. After the GPS treatment, the level of intracellular ROS level was increased, and DNA fragmentation and apoptotic morphology were observed. Liu et al. [41] investigated the effects of GPS on renal cell carcinoma (RCC) and its molecular mechanism for the first time. Studies concluded that the cell viability of 786-O and Caki-1 treated with gradient concentration of GPS (0, 75, 150, 300, 600 µg/mL) for 24 h showed a significant decrease. In addition, GPS induced the apoptosis of RCC cells through regulating the phosphoinositide 3-kinases/protein kinase B/mammalian target of rapamycin (PI3K/Akt/mTOR) signaling pathway.

#### 3.1.3. Inhibition of Cell Migration

Metastasis was believed to be one of the major challenges for a successful cancer treatment and the prevention of cancer. GPS exerted an inhibitory effect on cell migration in vitro in a dose-dependent manner. GPS could induce the microfilament network collapse and injure the cell shape and migration ability [30]. In addition, damulin B was reported to inhibit the migration of human lung cancer A549 and H1299 cells by down-regulating the protein production of matrix metalloproteinase-2 and -9 (MMP-2 and -9) [35].

#### 3.1.4. Regulation of Gut Microbiota

At present, accumulating evidence indicates that targeted intervention in the gut microbiota composition had shown encouraging results in cancer prevention and treatments. It was demonstrated that treatment with GPS significantly boosted the growth of short-chain fatty acids-producing bacteria that played an important role in maintaining the colonic health, and it down-regulated the relative abundance of sulfate-reducing bacteria, which were known to produce hydrogen sulfide and contribute to damage of the intestinal epithelium or even promote cancer progression in Apc^Min/+^ mice. Therefore, GPS exerted cancer-preventive effects and had the potential to be a new preventive medicine against colorectal cancer [42].

#### 3.1.5. Cell Cycle Arrest

Gypenoside LI, a gypenoside monomer from GP, had a potent cytotoxic effect on melanoma cells by arresting the cell cycle at the S phase and inhibiting melanoma cell proliferation through the up-regulation of miR-128-3p [14]. Studies had shown that GPS could induce G0/G1 arrest in A549 cells. The molecular mechanism of cell cycle arrest was investigated by Western blotting, showing that cyclin-dependent protein kinase inhibitors such as p16, p21, p27, and p53 proteins were accordingly increased with the increasing time of incubation with GPS in the A549 cells [20]. Flavonoids extracted from GP were reported to induce cycle arrest at both S and G2/M phases with the concurrent modulated expression of the cellular proteins cyclin A, B, p53, and p2, and then inhibit the proliferation of lung cancer A549 cells [43].

#### 3.1.6. Others

A growing amount of evidence had proven that senescence was an effective tumor-suppressive approach in cancer treatment. Gypenoside L could inhibit the proliferation of cancer cells by inducing senescence. Chronic administration of chemotherapeutic drugs resulted in the resistance of cancer cells due to protective autophagy. However, gypenoside L could render cancer cells more sensitive to chemotherapy such as cisplatin by impairing the protective autophagy. Based on this, gypenoside L could be explored as a promising agent for solving the problem of resistance [16].

#### 3.1.7. Toxicity on Normal Cells and Cancer Cells

Studies had shown that GPS was capable of exerting different alternative cytotoxicity in cancer cells and normal cells. Gypensapogenin H isolated from the hydrolysate of total saponins of GP exhibited potent growth inhibitory effects on tumor cells. It significantly inhibited the growth of human breast cancer cells, whereas it showed lower toxicity to normal human breast epithelial cells than the commonly used anti-cancer drugs paclitaxel and 5-Fu [36]. The above result was consistent with a study conducted to explore the cytotoxicity of GPS on normal peripheral blood mononuclear cells [29]. In addition, different cancer cells exerted different sensitivity to the sham gypenoside. The cytotoxic activities of gypsapogenin A isolated from the acid hydrolysate of total saponins against HepG2 and A549 cells were evaluated, and the results showed that the IC50 values of gypsapogenin A against HepG2 and A549 cells were 24.12 and 59.81 µM, respectively [44].

#### 3.1.8. Relationship between Anti-Cancer Activity and Structure of GPS

The anti-cancer activity of GPS was associated with its structure. Xing et al. [19] evaluated the cytotoxic activities of several dammarane saponins (gypenoside L, gypenoside LI, gypenoside LVI, and gypenoside XLVI) on A549 cells with ginsenoside as a positive control by CCK-8 assay and found that the loss of sugar might be related to the enhanced inhibition of cancer cell proliferation. Once the free hydroxyl in the C20 position was glycosylated, the anti-cancer activity was extremely decreased. As shown in Figure 5, gypenoside stereoisomers, gypenoside L (S configuration at C20), and gypenoside LI (R configuration at C20), both with free hydroxyls in the C20 position, showed stronger activity with IC50 values of 29.38 ± 2.52 µM and 21.36 ± 0.78 µM respectively against A549 cells compared with gypenoside LVI and gypenoside XLVI glycosylated in the C20 position. Cui et al. [45] evaluated the cytotoxic activities of damulin E and damulin F (their structures are shown in Figure 5) against A549 cells and showed that after being incubated for 24 h, compared with positive control (ginsenoside Rg3), damulin F showed stronger cytotoxic activity against A549 cells with the IC50 value of 19.8 ± 0.4 µM, whereas damulin E appeared to have weaker activity with the IC50 value of 38.9 ± 0.6 µM. Therefore, saponins with a double bond in C20 (21) exhibited stronger activities than saponins with a double bond in C20 (22). However, some studies [44,46,47,48] did not mention the IC50 value or positive control, and the cytotoxic activities could not compare, so the relationship of the structure and activities needed further investigation.

### 3.2. Anti-Atherogenic Effect

Accumulating evidences had demonstrated that GP could be developed as potential and promising agents for atherosclerosis (AS) treatment. The detailed parameters of experiments including experimental models, dosage, and possible mechanisms were listed in Table 2. The anti-atherogenic mechanisms are shown in Figure 6, which were mainly related to the reduction of foam cell formation, inflammation, DNA damage, and endothelial cells (ECs) apoptosis and the increasement of autophagy and cholesterol flux.

Inflammation occurred at different stages of AS progression. Therefore, anti-inflammatory therapy had become an important intervention to delay the progression of the disease. Recent studies had shown that the expression of TLR4 (Toll-like receptor 4) and its downstream signaling molecules MyD88 (Myeloid differentiation factor 88), NF-κB, and TNF-α in aortic tissue were inhibited after GP intervention (80 and 160 mg/kg for 10 weeks, i.g.), thus alleviating vascular inflammation [61]. Studies had demonstrated that phytoestrogens, compounds with structural similarities to estradiol, could bind weakly to estrogen receptors and play a major role in the prevention and treatment of AS. Gypenoside XVII (GP-17), a novel phytoestrogen, had a similar structure to estradiol, and its anti-atherogenic effect of GP-17 had been verified in vivo and vitro. GP-17 treatment (50 mg/kg) significantly decreased blood lipid levels, increased the expression of antioxidant enzymes, and decreased atherosclerotic lesion size, thus effectively preventing the progression of atherosclerosis in ApoE-/- mice. Oxidized low-density lipoprotein (Ox-LDL) in the pathogenesis of atherosclerosis was an exceptionally important risk factor, which contributed to endothelium apoptosis. GP-17 (50 µg/mL) significantly inhibited Ox-LDL-induced ROS generation and apoptosis in human umbilical vein endothelial cells (HUVECs) by the estrogen receptor alpha (ERα)/PI3K/Akt pathway. Furthermore, GP-17 had been verified as a selective ERα modulator for atherosclerosis treatment [49].

With the deepened understanding of the pathogenesis of AS, the inhibition of the formation of macrophage foam cells and atherosclerotic plaque had been paid more attention for new drug development. The formation of macrophage foam cells was through scavenger receptor-mediated endocytosis of Ox-LDL. CD36 receptor was the most important scavenger receptor in the macrophage cell membrane to mediate the endocytosis of Ox-LDL. Studies had shown that the expression of the scavenger receptor of CD36 in macrophages was downed by GPS intervention (100 µg/mL), whereby the intake of Ox-LDL in macrophages was reduced [50]. In addition, Shen et al. confirmed the inhibitory effect of gypenoside LVI (GPLVI) on Ox-LDL-induced foam cell formation for the first time. Treatment with GPLVI (25, 50, and 100 µg/mL) significantly promoted cholesterol efflux by increasing the expression of ATP-binding cassette transporter G1 (ABCG1), scavenger receptor class B type I (SRB1), and liver X receptor (LXR) α, thus reducing the formation of foam cells [52]. Several studies had shown that the activation of LXR could inhibit the development of atherosclerosis in mice, induce ATP-binding cassette transporter A1 (ABCA1) and apolipoprotein E gene expression, and increase cholesterol efflux, which lead to a decreased cholesterol burden in the arterial wall, as well as enhanced high-density lipoprotein (HDL) levels [63]. Gynosaponin TR1 was reported to be an agonist of LXR with selectivity for LXR-α over LXR-β, and this selectivity of TR1 toward LXR-α might prevent the elevation of triglycerides (TG) [63]. Recent studies had found that autophagy was closely related to AS, and moderate autophagy could prevent macrophages from transforming into foam cells to delay the occurrence and development of AS. Studies had shown that GPS (2.973 g/kg for 4 weeks, i.g.) might enhance the level of autophagy of AS model mice by regulating the mTOR/unc-51-like kinase 1 (ULK1) autophagy signaling pathway, alleviate the formation of atherosclerotic plaque, and prevent atherosclerosis [53].

### 3.3. Lipid-Regulating Effect

#### 3.3.1. Effect on Hyperlipidemia

A large number of studies had shown that GP had the effect of lowering blood lipid and was commonly used together with other traditional Chinese medicines or combined with Western medicines to treat hyperlipidemia. GPS was the major component that played an important role in lowering blood lipid levels. It was found that the nucleus structure of GPS was very similar to that of endogenous bile acids, which could activate the farnesoid X receptor in the liver, thus up-regulating the key enzymes CYP8B1 and CYP7A1 in bile acid synthesis, promoting the synthesis and secretion of bile acids, promoting lipid metabolism, and lowering blood lipid [64]. Mao [65] studied the lipid-lowering effect of gypenoside SL-1 and SL-2 isolated from GP on HepG2 cells in vitro. The results showed that SL-1 and SL-2 (10, 50, and 100 µg/mL) could decrease the content of TC, TG, and low-density lipoprotein (LDL) and increase the content of HDL, thus reducing lipid accumulation in HepG2 cells incubated with oleic acid. Currently, the first-line drug used for the treatment of hyperlipidemia was statins. However, its lipid-lowering effect could not reach the target proposed by the National Institutes of Health of the United States [66]. That was because the expression of proprotein convertase subtilisin/kexin 9 (PCSK9) while lowering lipids was enhanced, which further induced the increased low-density lipoprotein receptor (LDLR) degradation, thus inhibiting the lipid-lowering effect [67]. A number of clinical studies had reported that statin therapy might cause the increase in transaminase in the body while lowering lipid levels, thus affecting liver function [66]. Studies had demonstrated that GPS could significantly lower blood lipids, and when combined with simvastatin, the effect of lowering lipid levels was improved by suppressing the increased expression of PCSK9 and reducing the degradation of LDLR. In addition, the increase in serum transaminase induced by simvastatin was reversed, and liver function was improved [67,68]. Yin et al. isolated and identified ten novel saponins named yunnangypenosides (1–10) and a known one from GP. Furthermore, these compounds could significantly reduce lipid levels except for those of 2 [69].

#### 3.3.2. Anti-Obesity Effect

More and more evidence has shown that GP also possessed a potent anti-obesity effect. GP extract (GPE) (300 mg/kg) with much higher contents of gypenoside, gypenoside LI, and ginsenoside Rg3 exhibited a stronger anti-obesity effect than the positive group (Orlistat, 30 mg/kg). When GPE was given, the increases in body weight, fat mass, white adipose tissue, and adipocyte hypertrophy were suppressed, and the levels of serum triglyceride, total cholesterol, and low-density lipoprotein-cholesterol were lower than the high-fat-diet group. The lipid-lowering effect was associated with AMP-activated protein kinase (AMPK) activation, which led to increased SIRT1 expression [70]. GPE (300 mg/kg) named actiponin containing large amounts of damulin A and damulin B were successfully delivered to the skeletal muscle and liver, and it prevented or improved obesity in ob/ob mice by stimulating fatty acid oxidation and activating AMPK in these organs. Actiponin (200 mg/kg) reduced body weight and plasma total cholesterol level without any effect on food intake [71]. In addition, GPS could prevent high-fat diet-induced obesity by promoting energy expenditure. When treated with GPS (300 mg/kg), brown adipocyte tissue activity and white adipose tissue browning were increased [72]. Inhibiting adipogenesis was effective and promising for the treatment of obesity as well. An isolated novel saponin JS (100 µM) could inhibit adipocyte differentiation and adipogenesis, which might be associated with Wnt/β-catenin signaling activation [73]. Recently, a double-blind, randomized, clinical trial with a treatment duration of 16 weeks was conducted to assess the efficacy and safety of a commercially available capsule-form herbal supplement containing GPE on improving body composition in overweight males and females. Studies showed that GPE (450 mg/d) was capable of altering fat mass and fat distribution in overweight and obese males and females compared to a placebo. Following 16 weeks of treatment, a significant reduction in total body weight and total fat mass was observed. There was a significant difference between males and females. Males had a significant reduction in visceral fat and females had a significant reduction in gynoid fat [74]. It was proved that a high-fat diet can lead to changes of gut microbiota, which might further contribute to obesity. High ratio of Firmicutes to Bacteroidetes had been found in obese mice and human adults when compared to lean mice and human adults. GPS treatment (100 and 300 mg/kg) significantly decreased the ratio of Firmicutes to Bacteroidetes by 20% and 58.6% respectively and increased Akkermansia muciniphila abundance in the gut microbiota, thereby preventing the development of obesity [72]. At present, the application of GPS was mainly for internal use. GPS had an obvious lipid-lowering effect for external use as well, which could meet different needs aimed at the uneven distribution of fat in the body [75].

When considering these findings together, GP is a promising traditional Chinese medicine (TCM) used for hyperlipidemia and obesity.

### 3.4. Neuroprotective Effect

#### 3.4.1. Effect on Alzheimer’s Disease

Alzheimer’s disease (AD) is a common degenerative neurological disease, which is mainly manifested in the severe decline of cognitive function and memory. The characteristic pathological changes of AD include extracellular senile plaques formed by beta-amyloid protein (Aβ) deposition, neurofibrillary tangles formed by Tau protein hyperphosphorylation, and neuronal loss accompanied by glial cell proliferation. There were many theories about the pathogenesis of AD, including abnormal deposition of Aβ, Tau protein hyperphosphorylation, and so on [76]. At present, there was no specific cure drug, and only symptomatic treatment and therapeutics to delay the progression of AD were given to AD patients.

Aβ abnormal deposition was a crucial factor that drives AD pathogenesis, which was associated with neuroinflammation, oxidative stress, cell apoptosis and autophagy, etc. GPS could improve the learning and memory ability in dementia mice induced by D-galactose combined with sodium nitrite and aluminum trichloride, and the mechanism underlying might be related to the reduced oxidative damage and Aβ expression in brain tissues. Compared with the model group, the activity of SOD and GSH-Px were significantly increased, and the level of MDA was significantly decreased in brain tissues of the GPS group (250 mg/kg). In addition, the expression of Aβ42 decreased by treatment with GPS [77]. Under normal circumstances, microglial cells were in a static state, which might be activated into a classic activated state (M1 state) or alternative activated state (M2 state) when the body was in a pro-inflammatory environment for a long time. The former was harmful; in contrast, the latter was beneficial. A high level of Aβ in the brain could activate microglial cells to secrete pro-inflammatory factors and eventually develop into chronic inflammation. Treatment with 50 mg/L GPS for 24 h could attenuate Aβ-induced inflammation by reducing microglial activation and shifting microglial M1 to M2 state, and the process might be mediated by suppressors of cytokine signaling 1 (SOCS1) [78]. GPS also could improve cognitive dysfunction and chronic inflammation caused by injecting lipopolysaccharide in the hippocampus. GPS treatment (25, 50, and 100 mg/kg) for 21 days significantly decreased pro-inflammatory mediators such as IL-6, IL-1β, and NF-κB levels in the brain [79]. Pretreatment with GPS could effectively inhibit the Aβ_1-40_-induced increase in inducible nitric oxide synthase (iNOS) expression and cell apoptosis. Compared with Aβ_1-40_ group, the growth condition of cells incubated with GPS was improved obviously, the number of neurons was increased, and the synapses between adjacent neurons were more and compact [80]. Jia et al. investigated the neuroprotective effect against Aβ-induced cytotoxicity of a purified polysaccharide GPP1 in PC12 cells. The results showed that GPP1 exerted its protective effect via inhibiting oxidative stress and suppressing the mitochondrial apoptotic pathway [81]. Phytoestrogens had been proved to be protective against Aβ-induced neurotoxicity and regarded as a relatively safe agent for AD treatment. GP-17, as a novel phytoestrogen isolated from GP, had reported its neuroprotective effect [82,83]. GP-17 promoted the autophagy-based elimination of Aβ and prevented the formation of Aβ plaques in the hippocampus and cortex of APP/PS1 mice [82]. GP-17 exerted protective effect against Aβ_25-35_-induced neurotoxicity in nerve growth factor-induced differentiation of PC12 cells, including oxidative stress, apoptosis, and autophagic cell death, which might be related to estrogen receptor-dependent activation of PI3K/Akt pathways, inactivation of GSK-3β, and activation of nuclear factor-erythroid 2-related factor 2/antioxidant responsive element/heme oxygenase 1 (Nrf2/ARE/HO-1) pathways [83].

Most scholars believed that the damage of the cholinergic system in the brain was the main possible mechanism for the formation of AD, thus putting forward the “cholinergic hypothesis”. Cholinergic neurotransmitters were important for normal learning and memory. A decrease in cholinergic function, particularly within the basal forebrain, could result in a decline in memory and cognitive function with age. Scopolamine, an anticholinergic drug, caused memory impairments. The memory-enhancing effects of Gypenoside TN-2 and Gypenoside LXXIV (G-74) were investigated in scopolamine-treated mice in passive-avoidance and Morris water maze tests. Studies had demonstrated that TN-2 and G-74 potently ameliorated memory and learning impairment caused by scopolamine [84,85].

#### 3.4.2. Effect on Vascular Dementia

Vascular dementia (VD) was a clinical syndrome caused by various cerebrovascular disease and characterized by cognitive, learning, and memory impairments. VD was the second-most-common dementia in the elderly, after Alzheimer’s disease. With the aging of China’s population, the incidence of vascular dementia is increasing year by year. Large amounts of studies had demonstrated that the neurons injury or apoptosis were observed in the hippocampus of VD mice, which might be one of the important mechanisms of dementia. GPS had been reported to significantly improve learning and memory impairment and protect the neurons by inhibiting the expression of apoptosis-related protein P38 and caspase-3 [86,87]. It was found that the level of phosphorylated cAMP response element-binding protein (PCREB) in the hippocampus could affect the plasticity of synapses and the ability of learning and memory. The expression level of PCREB mRNA in the hippocampus of VD model group was significantly lower than that of sham operated group. By intraperitoneal injection of 100, 200, and 400 mg/kg GPS, the expression of PCREB was significantly up-regulated, suggesting that the decrease in hippocampal neuron damage and the improvement of learning and memory ability in VD mice might be related to the up-regulation of PCREB expression in hippocampal neurons [88]. In addition, the effect of GPS on improving cognitive function was closely associated with its antioxidant and anti-inflammatory activities. By GPS treatment (100 and 200 mg/kg for 61 days), the oxidative neuronal damage was ameliorated, and the activation of inflammatory astrocytes was reduced in the cortex and hippocampus after chronic cerebral hypoperfusion [89]. Together, GPS had therapeutic potential for the treatment of VD and might serve as a potential new anti-dementia agent.

#### 3.4.3. Anti-Parkinson’s Disease

Parkinson’s disease (PD) is a chronic neurodegenerative disease, which mainly occurs in middle-aged and elderly patients. The main clinical manifestation included bradykinesia, static tremor, autonomic nerve dysfunction, and myotonia. To date, large amounts of studies had demonstrated the anti-Parkinson’s property of GP, as shown in Table 3.

Studies on the pathogenesis of PD had demonstrated that the loss of dopaminergic neurons in the brain of PD patients was closely related to oxidative stress [96]. The 1-methyl-4-phenyl-1,2,3,6-tetrahydropyridine (MPTP)-induced decreases of glutathione content and SOD activity in the substantia nigra, which resulted in oxidative stress, loss of nigral dopaminergic neurons, and motor dysfunction, were all attenuated by treatment with GPS [90]. In vitro, GPS protected primary cultured dopaminergic neurons against 1-methyl-4-phenylpyridinium ion (MPP+)-induced oxidative injury as well. It was found that GPS pretreatment had stronger protective effect than GPS co-treatment or post-treatment [91]. L-3, 4-Dihydroxyphenylalanine (L-DOPA) was the natural precursor of dopamine and was the most effective therapy for PD. However, chronic L-DOPA administration resulted in a loss of drug efficacy and irreversible adverse effects including motor and non-motor dysfunctions in both patients with PD and animal models of PD [92,93]. Several studies had reported that GP could ameliorate L-DOPA-induced adverse effects and might be developed as a useful agent for adjuvant therapeutics of PD receiving long-term treatment with L-DOPA. GPS (25 and 50 mg/kg) and GPE (50 mg/kg) could effectively attenuate the development of L-DOPA-induced-dyskinesia without compromising the anti-Parkinson effects of L-DOPA by reducing ΔFosB expression and ERK1/2 phosphorylation in the 6-hydroxydopamine (6-OHDA)-lesioned rat model of PD [92]. Studies showed that MPTP-lesioned mice exhibited deficits associated with habit learning and spatial memory, which were further aggravated by treatment with L-DOPA (25 mg/kg). However, treatment with GPS (50 mg/kg) ameliorated memory deficits via increasing the activation of the dopaminergic neuronal system and modulating the phosphorylation of the N-methyl-D-aspartate (NMDA) receptor-mediated signaling system [93]. In addition, GPS (50 mg/kg) and GPE (50 mg/kg) improved the symptoms of anxiety disorders in the MPTP-lesioned rat model of PD with or without L-DOPA treatment [94].

### 3.5. Effects on Ischemia–Reperfusion(I/R) Injury

Tissue ischemia–reperfusion (I/R) injury is the damage caused by excessive free radical attacks on cells in tissues after blood flow recovery on the basis of ischemia. Numerous evidence had confirmed that the saponins and flavonoids in GP exhibited the improvement effect on tissue I/R injury, including renal, hepatic, and myocardial I/R injury (as shown in Table 4).

It had been demonstrated that the generation of I/R injuries is closely associated with inflammation, oxidative damage, and cell apoptosis. Studies in vitro and in vivo had shown that the effect of GPS on I/R-induced injury was mainly attributed to its anti-inflammation, anti-oxidative, and anti-apoptotic activity. An oral treatment with 50 mg/kg GP 1 h prior to ischemia effectively attenuated the I/R-induced increase in the levels of hepatic lipid peroxidation and serum alanine aminotransferase and suppressed the I/R-induced increase in the pro-apoptotic protein levels of Bax and cytochrome c and the activity of caspase-3/8, as well as the I/R-induced decrease in the levels of anti-apoptotic protein Bcl-2 [102]. Ye et al. [101] evaluated the protective effect against I/R-renal injury and its underlying molecular mechanisms. Pretreatment with 50 mg/kg GP significantly inhibited I/R-induced up-regulation of concentrations of serum creatinine and urea nitrogen. Furthermore, pretreatment with GP inhibited the I/R-induced production of pro-inflammatory cytokines, oxidative damage, and apoptosis. In myocardial I/R injury, GPS was shown to reduce infarct size and alleviate the impairments in cardiac structure and function, which was associated with the preservation of mitochondrial function and reduction of oxidative stress in the myocardial cells [98]. In addition, studies had shown that gypenoside had strong capacity to inhibit ER stress-induced cardiomyocyte apoptosis through the blockade of C/EBP-homologous protein (CHOP) pathway and activation of PI3K/Akt pathway [100]. A study by Yu et al. showed that GP protected cardiomyocytes against I/R injury by inhibiting NF-κB activation via the MAPK signaling pathway [97]. Another study by Chang et al. confirmed that gypenoside A functioned its protective effect on myocardial I/R injuries by suppressing miR-143-3p level via the activation of AMPK/FOXO1 pathway [99]. These studies indicated that the effects of GPS on I/R injuries could be multi-pronged.

### 3.6. Hepatoprotective Effects

At present, numerous studies had demonstrated that GP had the strong protective property for various kinds of liver injury (as shown in Table 5).

#### 3.6.1. CCl_4_-Induced Liver Injury

Researchers reported that GPP possessed a protective effect against hepatic injury in rats caused by carbon tetrachloride (CCl_4_). GPP (40 and 80 g/kg for 30 days, i.g.) could stabilize the liver cell membrane, effectively prevent the leakage of substance in cytoplasm, and reduce the level of serum transaminase. In addition, the up-regulated expression of iNOS mRNA in liver tissues of a rat model, which led to the production of nitric oxide (NO) and further induced liver injury, was attenuated following GPP treatment [103]. CCl_4_ was metabolized by cytochrome P-450 in the liver endoplasmic reticulum to produce a free radical that could lead to lipid peroxidation, liver damage, and necrosis. GPS exhibited a potent protective effect on CCl_4_-induced acute and chronic liver damage, which was attributed to its free radical scavenging capacity [104,105]. In addition, GPS was reported to also possess anti-fibrotic potential on CCl_4_-induced liver fibrosis in vivo and vitro studies, which supported its clinical use in alleviating the progression of hepatic fibrosis. GPS could improve the collagenolytic activity, decrease the deposition of collagen in the liver, promote the proliferation of liver cells, and thus suppress the onset and enhance the recovery of liver fibrosis induced by CCl_4_ in rats [104]. Chen et al. [106] investigated the effects and mechanisms of GPS on ameliorating liver fibrosis induced by CCl_4_ and found that the transforming growth factor-β (TGF-β) signaling was inhibited by treatment of GPS, consequently suppressing the proliferation of hepatic progenitor cells and inhibiting their differentiation into myofibroblasts.

#### 3.6.2. Exhaustion Exercise-Induced Liver Injury

Generally, extreme exercise leads to the generation of a large number of free radicals, and when the generated free radicals exceed the scavenging capacity of the body’s anti-oxidation defense system, the body is in the state of oxidative stress and further induces cell apoptosis. It was reported that GPP possessed protective effects on exhaustion exercise-induced hepatocyte apoptosis by its anti-oxidative and anti-apoptotic activities. GPP administration decreased the expression of Bax and increased the expression of Bcl-2. Meanwhile, GPP protected liver tissue against oxidative damage induced by exhaustion exercise as evidenced by significant increase in the SOD, GSH-Px, and CAT activities and decrease in the content of MDA [107].

#### 3.6.3. Alcoholic Liver Injury

Studies on alcoholic liver injury indicated that oxidative stress was one of the most important mechanisms, which further led to oxidative damage and inflammatory response. GPS exerted the protective effect on alcoholic liver injury by activating the Nrf2 signaling pathway to improve the level of antioxidant defense system and inhibiting the nuclear transposition of NF-κB to reduce the inflammatory response [108].

#### 3.6.4. High Choline-Induced Liver Injury

The dietary ingestion of high choline was highly linked to liver oxidative stress injury. GPS was reported to exert the significant protection against liver oxidative stress injury in high choline-treated mice [109].

#### 3.6.5. Fatty Liver Disease

GPS was believed to be a promising agent for the prevention of fatty liver disease. The protective effects of GPS on fatty liver disease were investigated in rats treated with high-fat and cholesterol diet and alcohol. GPS could prevent liver fatty degeneration through the up-regulation of PPAR-α expression in the liver to inhibit lipid peroxidation and hepatocyte apoptosis, thus ameliorating hepatic steatosis and mitochondrial damage and improving hepatic function [110]. GP was reported to have a hepatoprotective effect on nonalcoholic fatty liver disease (NAFLD) as well. Studies had shown that GP could alleviate NAFLD with following mechanisms: (1) reduction of inflammatory response by regulating the balance of liver Treg/Th17 cells, reducing the production of pro-inflammatory factors, and increasing the production of anti-inflammatory factors, so as to reduce liver inflammatory response and protect liver tissue; (2) reduction of oxidative stress as evidenced by increased SIRT6 and phase 2 anti-oxidant enzyme expression in liver tissues; (3) structural alterations of gut microbiota [111,112,113].

### 3.7. Hypoglycemic Effect

Diabetes mellitus is a chronic metabolic disease caused by genetic and environmental factors for a long time, which has become the third category of diseases seriously endangering human health. At present, the existing oral hypoglycemic drugs in China, such as metformin and sulfonylureas, could cause serious adverse effects [114]. The long-term use of insulin will lead to decreased insulin receptor sensitivity of the body, insulin resistance, and finally aggravate the disease. Therefore, it was urgent to seek new, safe, and effective oral hypoglycemic drugs. In recent years, natural medicine has attracted more and more attention because of its long-lasting effect and fewer side effects. Researchers had made great progress in searching for anti-diabetic components from traditional Chinese medicine resources, including GP. At present, the research on anti-diabetes mainly focused on GPS, GPP, and crude extracts (Table 6) [115].

GPS, as the main active ingredient, had been reported for its hypoglycemic effects. Phanoside, a novel dammarane-type saponin with a molecular mass of 914.5 Da, was isolated from ethanol extract, and four stereoisomers differing in configurations at positions 21 and 23 were identified, each of which was found to stimulate insulin release from isolated rat pancreatic islets [122]. Phanoside-induced insulin release might be mediated via the K-ATP channel and L-type Ca^2+^ channel [123]. However, it could dose-dependently stimulate insulin release at both low and high glucose levels (3.3 mM and 16.7 mM), suggesting that its insulin secretory effect was not glucose-dependent, which could potentially induce severe hypoglycemia and was similar to that of the well-known antidiabetic drug sulfonylurea. Gylongiposide I, which was identified in screening the active components of GP, was different from Phanoside and displayed unique abilities to stimulate insulin release at high glucose levels (16.7 mM) but limited effects at a low glucose concentration (3.3 mM) [124]. In vitro studies had shown [125] that GP exerted stronger inhibitory α-glucosidase activity with IC50 (42.8 µg/mL) as compared with acarbose (IC50 at 53.9 µg/mL). Based on this, GP could be developed as a novel anti-diabetic drug taken to reduce postprandial blood glucose. Compared with the traditional oral hypoglycemic drugs that had no obvious protective effect on the kidney, GPS displayed a protective effect on the kidney while lowering blood glucose a well as significantly improving renal tubule congestion and renal tubule epithelial edema in diabetic mice [116]. AMPK was a key regulator of glucose and lipid metabolism. Activated AMPK decreases blood glucose levels by increasing the glucose uptake and glycogen synthesis, stimulating the translocation of glucose transporter 4 (GluT4) to the cell membrane as well as up-regulating GluT4 expression. Therefore, AMPK activators were considered promising candidates for type-2 diabetes mellitus. Two novel dammarane-type saponins named damulin A and B isolated from GP could stimulate glucose uptake via AMPK activation in L6 myotube cells [117]. Protein tyrosine phosphatase 1B (PTP1B), a negative regulator of the insulin-signaling pathway, could inhibit or terminate the insulin signaling pathway transduction and further regulate negatively insulin sensitivity. Therefore, inhibition of PTP1B was an effective therapeutic approach to treatment type-2 diabetes mellitus. In a recent study conducted by Wang et al. that aimed to find the hypoglycemic active components in GP, 12 compounds and their hydrolysis products were isolated and identified from GPS, and 11 compounds exerted inhibitory activity against PTP1B [126]; these results were in line with previous studies [127]. It was reported that insulin sensitivity could be affected by inflammation. Inflammatory factors such as TNF-α and IL-6 produced by nonspecific inflammation could interfere with the conduction of insulin signaling pathway and thereby lead to decreased insulin sensitivity. Studies showed that insulin sensitivity in diabetic rats treated with GPS was improved by blocking the NF-κB signaling pathway and reducing the secretion of inflammatory factors. However, GPS had no influence on the streptozotocin-induced rat islet cells damage [128].

A large number of studies proved that GPP had hypoglycemic effect. Du et al. [119] established a model of type 2 diabetic rats by a high-fat and high-sugar diet combined with a small dose of streptozotocin to study the effects of GPP on blood glucose and its possible mechanism. The results showed that the fasting blood glucose and glucose tolerance were reduced by the treatment with GPP. Moreover, the fasting serum insulin level was increased in a dose-dependent manner, suggesting that the hypoglycemic effect of GPP was related to the secretion of insulin. In vitro studies showed that GPP had a strong inhibitory effect on α-amylase in a dose-dependent manner, which was equivalent to about 50% of the inhibitory effect of acarbose at the same concentration [118].

The ethanol extract of GP grown in Hainan could significantly reduce the blood glucose level of alloxan-induced hyperglycemia mice. There was no statistically difference between the medium-dose group (2 g/kg) and the metformin group; the high-dose group (4 g/kg) showed a stronger hypoglycemic effect than the metformin group. More importantly, it could significantly inhibit the increase in blood glucose induced by glucose, but it exhibited no obvious effect on the blood glucose of normal mice [120]. Huang et al. [129] estimated the effects of different components of GP on blood biochemical indexes of hyperglycemic mice and found that both GPE and 90% ethanol elution parts via macroporous resin chromatography could reduce the blood glucose of diabetic mice and alleviate the symptoms of “three more and one less”. Yassin et al. [121] explored the effect of GPE (1600 mg/kg/d) on hepatic glucose output (HGO) in spontaneously type 2 diabetic Goto-Kakizaki rats, and the results showed that the hepatic insulin sensitivity was improved, while HGO and plasma glucose levels were reduced. This was in agreement with results from a clinic trial conducted to evaluate the anti-diabetic effect of GP tea in drug-naïve type 2 diabetic patients [130,131]. Moreover, the biosecurity of tea was proved clinically, as no hepatoxicity and nephrotoxicity or other adverse effects such as hypoglycemia were observed in the trial. In addition, the antidiabetic effect of GP was associated with the stimulation of insulin release from the islets. An oral treatment of GP water extract (0.3 g/kg of body weight daily) for two weeks in GK rats improved glucose tolerance and stimulated insulin release. GP-induced insulin release was partly mediated via K-ATP and L-type Ca^2+^ channels [132].

### 3.8. Others

In addition to the above significant biological activities, GP also possessed an additional anti-aging effect, anxiolytic effect, anti-gastric ulcer effect, anti-fatigue effect, anti-platelet aggregation activity, antidepressant effect, sedative and hypnotic effect, immunomodulatory effect, antiviral activity, antimicrobial activity, and effects on osteoporosis and hyperuricemia, as Table 7 shown.

## 4. Conclusions and Perspectives

*G. pentaphyllum* (Thunb.) Makino is a highly promising TCM that has attracted great attention with a wide range of potent bioactivities, including hypoglycemic, anti-tumor, hepatoprotective, neuroprotective, and so on. As a typical edible and medicinal plant, GP has been developed in the field of animal feed to improve the quality of meat and has a pretty good market prospect. However, GP was widely distributed and rich in natural resources. The chemical composition of GP has not been studied thoroughly. Most research about its pharmacological properties were only limited to GPS. Thereby, further pharmacological research on single gypenoside and flavonoids are required to extend our understanding of pharmacological properties. Another important issue was that the underlying mechanism of bioactivities was still inadequately researched. In the future, the new technologies should be explored to deepen our understanding of GP, which can provide the basis for clinical drug use and preparation development.

## Figures and Tables

**Figure 1 molecules-26-06249-f001:**
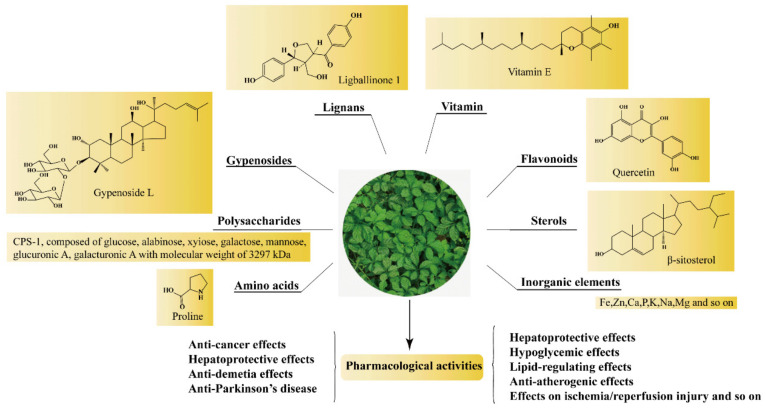
The main active components and pharmacological activities of GP.

**Figure 2 molecules-26-06249-f002:**
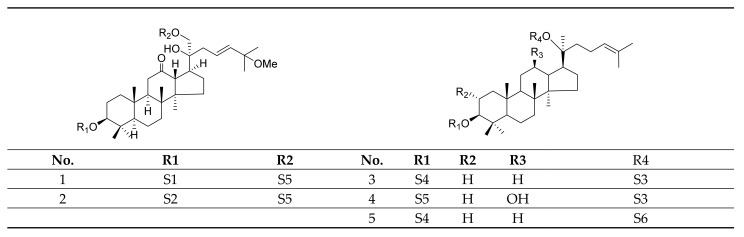
Structures of new dammarane-type saponins.

**Figure 3 molecules-26-06249-f003:**
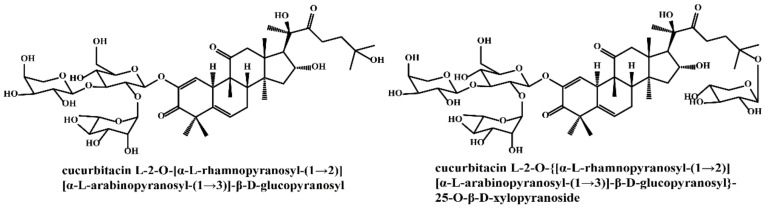
Structures of two cucurbitane-type saponins.

**Figure 4 molecules-26-06249-f004:**
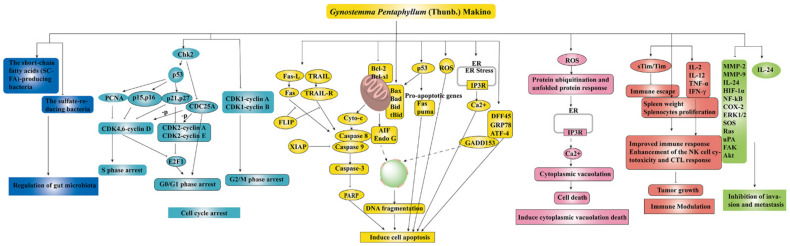
The anti-cancer mechanisms of GP ↑ meant up-regulation, ⊥ meant down-regulation.

**Figure 5 molecules-26-06249-f005:**
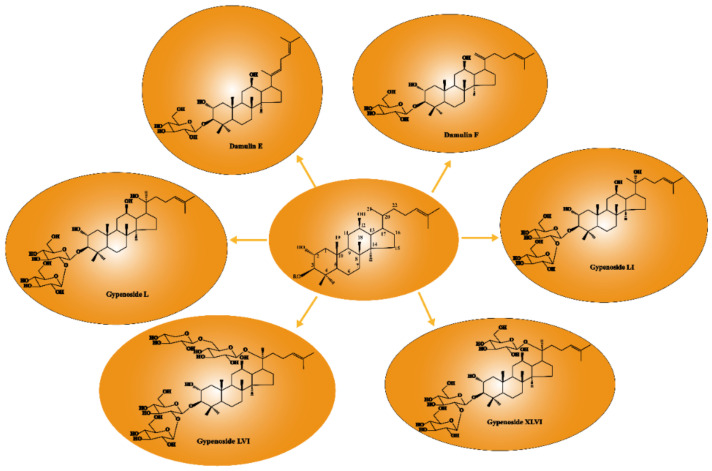
Relationship between anti-cancer activity and structure of GPS.

**Figure 6 molecules-26-06249-f006:**
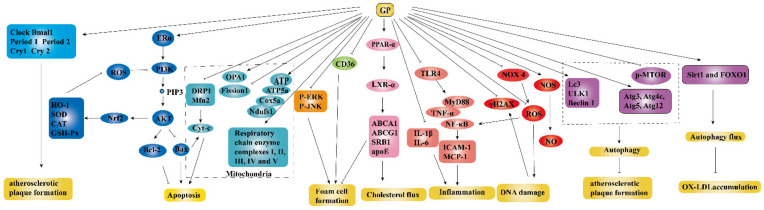
Anti-atherogenic effect of GP ↓ meant up-regulation, ⊥ meant down-regulation.

**Table 1 molecules-26-06249-t001:** Anti-cancer effect of GP.

Active Ingredient	Model	Mechanism	Ref.
Gypenoside LI	Melanoma cells (A375 and SK-MEL-28)	Induced intrinsic apoptosis along with S phase arrest, decreased the levels of CDK2, P-CDK2, Bcl-2, and FLIP, increased the levels of caspase 9, PARP cleavage, and BID death agonist, and increased the level of tumor suppressor miR-128-3p.	[14]
Gypenoside LI	Human breast cancer cells (MDA-MB-231 and MCF-7)	Inhibited the proliferation by decreasing the expression of ERCC6L, down-regulated the expression of MMP2 and MMP9, suppressed the migration ability of breast cancer cells, induced cell apoptosis, increased cytochrome c release from mitochondria into the cytoplasm, increased levels of Bax, decreased levels of PARP-1 and Bcl-2, induced cell cycle arrest in G0/G1 phase via down-regulating E2F1 and decreasing CDK2, CDK4, and cyclin D1.	[15]
Gypenoside L	Human hepatic cancer HepG2 cells and esophagus cancer ECA-109 cells	Caused cell cycle arrest at S phase, reduced the expression of CDK2, CDK4, CDK6, and cyclin D1, activated senescence-related cell cycle inhibitor proteins (p21 and p27) and their upstream regulators, activated p38 and ERK MAPK pathways and NF-κB pathway to induce senescence.	[16]
Gypenoside L	Human esophageal cancer cells (ECA-109 and TE-1)	Induced cell death, which was associated with lysosomal swelling and autophagic flux inhibition, increased the levels of ROS, triggered protein ubiquitination and ER stress response, leading to Ca^2+^ release from ER IP3R-operated stores and finally cell death.	[17]
Gypenoside L	Human hepatocellular carcinoma cells	Increased the intracellular ROS levels, which triggered protein ubiquitination and unfolded protein response, resulting in Ca^2+^ release from ER IP3R-operated stores and finally cytoplasmic vacuolation and cell death.	[18]
Gypenoside L and Gypenoside LI	Human lung cancer A549 cells	Inhibited A549 cell migration from the upper to the lower chamber through the membrane, decreased the expression of MMP2 and MMP9, gypenoside L down-regulated the expression of CDK2 and CDK4 proteins rather than CDK1 protein, while gypenoside LI suppressed the expression of CDK1 protein rather than CDK2 and CDK4 proteins. Gypenoside L induced G0/G1 arrest and gypenoside LI induced G2/M arrest in A549 cells; induced A549 cell apoptosis through extrinsic pathway and intrinsic pathway, boosted the production of ROS, led to more cytochrome c release from mitochondria into cytoplasm and less expression of procaspase 8.	[19]
GPS	Human lung cancer A549 cells	Increased p16, p21, p27, and p53 proteins and induced G0/G1 arrest; induced apoptosis by up-regulating Bax, caspase-3, and caspase-9 and down-regulating Bcl-2 levels.	[20]
GPS	Human colon cancer 205 cells	Induced DNA fragmentation and cell apoptosis, induced ROS and Ca^2+^ production, decreased the expression of Bcl-2 and Bcl-xl but increased the expression of Bax, increased the levels of p53 and promoted the release of cytochrome c and the activation of caspase-3, induced G0/G1 arrest, decreased the levels of cyclin D3, cyclin E, CDK4, CDK6 and CDK2, increased the levels of p15, p16, p21, p27, and p53.	[21]
GPS	Human tongue cancer SCC-4 cells	Induced ER stress and the production of reactive oxygen species and Ca^2+^, changed the ratio of Bcl-2 and Bax, followed by the dysfunction of mitochondria, caused cytochrome c release, activation of caspase-3; decreased the levels of cyclin D2 and cyclin E, increased the levels of Chk2, p53, p16, and p21, which led to G0/G1 arrest.	[22]
GPS	Human lung carcinoma A549 cells	Induced an arrest at both S phase and G2/M phase, increased the expression of cyclin E and PCNA, decreased the expression of cyclin A and B; induced apoptosis, decreased the expression of Bcl-2 and increased the expression of Bax, activated caspase-3 and the downstream substrates, DFF45 and PARP-1.	[23]
GPS	Human oral squamous carcinoma SAS cells	Inhibited the metastatic and invasive capacity of oral cancer cells, decreased the abundance of several proteins, including NF-κB, COX-2, ERK1/2, MMP-9, MMP-2, SOS, Ras, uPA, FAK, and Akt, decreased mRNA levels of MMP-2, MMP-7, and MMP-9.	[24]
GPS	Human oral cancer SAS cells	Reduced the levels of CDC25A, cyclin E, cyclin A, and CDK2, increased the levels of p53 and p21, caused G0/G1 phase arrest; triggered apoptotic cell death, increased levels of Bax but inhibited the levels of Bcl-2 and Bcl-xl, stimulated the release of cytochrome c, AIF, and Endo G, increased the translocation of AIF, Endo G, and GADD153, increased protein levels of puma, CAD, caspase-9, and caspase-3, TRAIL, Fas, FasL, ATF-4, and GRP78 but reduced levels of XIAP.	[25]
GPS	Human oral cancer HSC-3 cells	Decreased the depolarization of mitochondrial membrane potential, induced cell death, G2/M phase arrest, and apoptosis, altered gene expression such as the expression of GTP binding protein.	[26]
GPS	Human hepatocellular carcinoma HepG2 cells	Inhibited HIF-1α mRNA expression, as well as disturbing HepG2 migration and invasion.	[27]
GPS	Human hepatocellular carcinoma HepG2 cells	Up-regulated IP3R and SOC proteins (STIM1 and Orai1) and down-regulated SERCA protein, which led to increased extracellular Ca^2+^ influx and ER Ca^2+^ release and decreased ER Ca^2+^ uptake. This process perturbed intracellular calcium homeostasis and finally triggered Ca^2+^-dependent apoptosis.	[28]
GPS	Human colon cancer SW620 cells and Human esophageal cancer Eca-109 cells	Caused cell membrane integrity damage, promoted ROS production, decreased ∆φ_m_ level, induced apoptotic morphology such as cell shrinkage and chromatin condensation, initiated apoptotic response, inhibited cell migration.	[29]
GPS	Human colon cancer SW-480 cells	Increased the plasma membrane permeability of SW-480 cells, decreased the ∆φ_m_ level significantly, increased the level of intracellular ROS level, produced DNA fragmentation and induced apoptosis, caused serious microfilament network collapse as well as the significant decrease in the number of microvilli.	[30]
GPS and flavonoids	Prostate cancer PC-3 cells	Induced both S and G2/M phases arrest and cell apoptosis, increased the expression of Bad, Bax, and caspase 3, decreased the level of Bcl-2 and Bcl-xl, modulated the expression of G2 and M checkpoint regulators, cyclin A and B.	[31]
A neutral polysaccharide fraction (CGPP)	H22 tumor-bearing mice	Reduced the tumor weight, promoted splenocytes proliferation, promoted cytokine secretion (IL-2, TNF-α, IFN-γ), enhanced the NK cell cytotoxicity and CTL response.	[32]
Acidic polysaccharides (GP-B1)	B16 tumor-bearing mice	Inhibited the tumor growth and improved the immune organ status, increased the relative weight of spleen and increased splenocytes proliferation, increased serum TNF-α, IFN-γ, and IL-12 level, reduced IL-10 level.	[33]
A nonpolar fraction (EA1.3A)	Human breast adenocarcinoma MDA-MB-453 cells	Inhibited the cell growth and decreased colony formation ability, induced G0/G1 arrest and apoptosis.	[34]
Damulin B	Human lung carcinoma cells (A549 and H1299)	Decreased the level of Bcl-2, up-regulated the pro-apoptotic proteins Bax, Bid, and tBid, led to the release of cytochrome c in the cytoplasm, activated the extrinsic protein procaspase-8, cleaved caspase-8 emerged, and reduced the intrinsic protein procaspase-9, initiated cell apoptosis; up-regulated the expression of p53, suppressed the expressions of CDK4, CDK6, and cyclin D1, thus blocking the cell cycle at early G0/G1 phase; inhibited the expression of MMP-2 and MMP-9, up-regulated the expression of IL-24 protein, inhibited cell migration.	[35]
Gypensapogenin H	Human breast cancer MDAMB-231 cells	Decreased the expression of cyclin D1, E2F1, CDK4, and CDK2, increased p21 expression, blocked the cell cycle; down-regulated the anti-apoptotic protein Bcl-2, increased the levels of Bax, cytochrome c, cleaved caspase-3, cleaved caspase-9, and P-PARP, initiated cell apoptosis.	[36]
	Human prostate cancer cells (DU145 and 22RV-1)	Decreased the expression of cyclin D1 and CDK4, increased the expression of p21 to induce cell cycle arrest; decreased anti-apoptotic Bcl-2 protein while Bax, cleaved caspase-3 and -9 increased.	[37]

NF-kB, Nuclear factor-kappa B; COX-2, Cyclooxygenase-2; ERK, Extracellular signal-regulated kinase; SOS, Sevenless homolog; uPA, Urokinase-type plasminogen activator; FAK, Focal adhesion kinase; PCNA, Proliferating cell nuclear antigen; PARP, Poly ADP-ribose polymerase; DFF45, DNA degradation factor 45 KD; ER, Endoplasm reticulum; HIF-1α, Hypoxia-inducible factor-1α; Oria1, Calcium release-activated calcium channel protein 1; STIM1, Stromal interacting molecule 1; SOC, Store-operated Ca^2+^ channel; SERCA, Sarco/endoplasmic-reticulum-Ca^2+^-ATPase; CTL, Cytotoxic T lymphocytes; ERCC6L, Excision repair cross-complementation group 6 like; BID, BH3-interacting domain; ATF-4, Activating transcription factor 4; GRP78, Glucose-regulated protein 78 KD; CDK, Cyclin-dependent kinases; E2F1, E2F transcription factor 1; IL, Interleukin; TNF, Tumor necrosis factor; IFN, Interferon; IP3R, Inositol triphosphate receptor; FLIP, FLICE inhibitory protein; MAPK, Mitogen-activated protein kinase; Chk2, Checkpoint kinase 2; AIF, Apoptosis-inducing factor; CDC25A, Cell division cycle protein 25A; XIAP, X-linked inhibitor of apoptosis protein; TRAIL, TNF-related apoptosis-inducing ligand; GADD153, Growth arrest and DNA damage-inducible 153; GTP, Guanosine triphosphate.

**Table 2 molecules-26-06249-t002:** Anti-atherogenic effect of GP.

Active Composition	Models	Dosage	Activity/Mechanism	Ref.
Gypenoside XVII (GP-17)	A high-fat diet-induced AS model in ApoE-/- mice	50 mg/kg for 10 weeks, i.g.	Decreased blood lipid levels, increased the expression of antioxidant enzymes (SOD, GSH-Px, CAT) and decreased the MDA level in serum, decreased atherosclerotic lesion size.	[49]
	Ox-LDL-induced HUVECs injury model	6.25, 12, 25, 50, 100 µg/mL	Reduced the ROS generation, protected HUVECs against Ox-LDL-induced apoptosis and oxidative stress, increased the expression of ERα, alleviated atherosclerosis via the ERα-mediated PI3K/Akt pathway.	
GPS	Ox-LDL-induced foam cell formation in THP-1 macrophage	40–200 µg/mL	Down-regulated the expression of the receptor of CD36 in macrophages, up-regulated the expression of the receptor of ABCA1, LXR-α, PPAR-α in macrophages, then reduced the intake of Ox-LDL in macrophages, andpromoted the efflux of Intracellular cholesterol.	[50]
GPS	A high-fat diet-induced AS model in ApoE-/- mice	200 mg/kg and 400 mg/kg for 8 weeks, i.g.	Lowered the level of TC, TG and LDL-C, inhibited the formation of vascular wall hyperplasia, increased the expression of PPAR-γ, LXR-α and ABCA1. It might promote cholesterol efflux through activation of PPAR-γ/LXR-α/ABCA1 signaling pathway.	[51]
Gypenoside LVI (GPLVI)	Ox-LDL-induced foam cell formation in RAW264.7 cells	25, 50 and 100 µg/mL	Inhibited Ox-LDL-induced foam cell formation, promoted cholesterol efflux, inhibited inflammatory response, induced ABCG1, SRB1 and LXRα expression, blocked phosphorylation of ERK and JNK.	[52]
GPS	A high-fat diet-induced AS model in ApoE-/- mice	2.973 g/kg for 4 weeks, i.g.	Lowered the levels of TG, TC and LDL-C, elevated HDL-C level, reduced atheromatous plaques in aortic canal, increased the expression of ULK1, Beclin1 and LC3, decreased p-mTOR expression, relieved the formation of atherosclerotic plaque and prevented atherosclerosis possibly through regulating the autophagy.	[53]
GPS	A high-fat diet-induced AS model in SD mice	40, 80, 160 mg/kg for 7 weeks, i.g.	Inhibited the expression of ICAM-1, MCP-1 and NF-κB P56, delayed the process of atherosclerosis.	[54]
GPS	A high-fat diet-induced AS model in ApoE-/- mice	2.973 mg/kg for 7 weeks, p.o.	Enhanced PI3K and p-Akt expression and down-regulated expression of p-Bad, cytochrome c, cleaved caspase 3, cleaved caspase 9, down-regulated DRP1 and Mfn2 protein levels, inhibited the formation of atherosclerosis via the PI3K/Akt/Bad pathway.	[55]
GPS	Cholesterol-induced DNA damage in HUVECs	1, 10 and 100 µg/mL	Decreased the expression of NOX4 and inhibited ROS production, increased the activity of NOS, decreased the γH2AX expression, alleviated DNA damage.	[56]
GPS	Ox-LDL-induced foam cell formation in THP-1 macrophage	5, 10 and 25 µM	Increased the LC3-II levels, decreased p62 expression, increased the autophagosome puncta numbers, increased SIRT1 and FOXO1 levels, rescued the impaired autophagy flux, reduced the accumulation of Ox-LDL and foam cell formation, prompted lysosome biogenesis and restored its function.	[57]
GPS	A high-fat diet-induced AS model in ApoE-/- mice	2.973 g/kg for 4 weeks, i.g.	Increased the expression of Clock, Bmal1, Period 1, Period 2, Cry1 and Cry 2 proteins, alleviated the formation of atherosclerotic plaque.	[58]
Gypenoside A	Ox-LDL-induced injury in ECs (EA.hy926)	100 µg/mL	Up-regulated the expression of OPA1 and down-regulated the expression of Fission1, increased ATP level, increased the activity of respiratory chain enzyme complexes I, II, III, IV and V, protected ECs by affecting mitochondrial energy metabolism and fusion lysis.	[59]
GPS	A high-fat diet-induced AS model in ApoE-/- mice	2.973 g/kg for 4 weeks, i.g.	Increased the expression of autophagosome related proteins Atg3, Atg4c, Atg5, Atg12, promoted the formation of autophagosome, reduced the serum lipid level.	[60]
GPS	AS model established by a high-fat diet (HFD) and vitamin D3 injection in SD mice	40, 80, 160 mg/kg for 10 weeks, i.g.	Inhibited the expression of TLR4, MyD88, NF-κB, TNF-α, alleviated the inflammatory response.	[61]
GPS	Lipopolysaccharide-induced inflammatory injury in HUVECs	10, 50, 100 and 200 µg/mL	Inhibited the activation of NF-κB via ROS scavenging and reduced the production of inflammatory cytokines.	[62]

PPAR, Peroxisome proliferator-activated receptor; LDL-C, Low-density lipoprotein cholesterol; TC, Total cholesterol; HDL-C, High-density lipoprotein cholesterol; JNK, c-Jun N-terminal kinase; DRP-1, Dynamin-related protein 1; Mfn2, Mitofusin 2; NOX4, Nicotinamide adenine dinucleotide phosphate-oxidase 4; γH2AX, Phosphorylation of histone H2AX; NOS, Nitric oxide synthase; FOXO1, Forkhead Box O1; Cry, Crytochrome; OPA1, Optic atrophy protein 1; SOD, Superoxide dismutase; MDA, Malondialdehyde; GSH-Px, Glutathione peroxidase; CAT, Catalase; LDL-C, Low-density lipoprotein cholesterol; ICAM-1, Intercellular cell adhesion molecule-1; MCP-1, Monocyte chemotactic protein 1; SIRT1, Silent information regulator 1; LC3, Microtubule-associated protein 1 light chain 3; Atg, Autophagy-related gene.

**Table 3 molecules-26-06249-t003:** Anti-Parkinson’s disease effect of GP.

Models	Medicinal Composition	Dosage	Activity/Mechanism	Ref.
MPTP-induced rat model of PD	GPS	100, 200, and 400 mg/kg, i.p.	Attenuated the motor deficits and striatal dopamine loss in a dose-dependent manner, increased the number of tyrosine hydrolase (TH)-immunopositive neurons, increased glutathione content, and enhanced SOD activity in the substantia nigra.	[90]
MPP^+^-induced oxidative injury of dopaminergic neurons in primary nigral culture	GPS	50, 100, 200, 400 µg/mL	Increased glutathione content, enhanced activity of GSH-Px and SOD, attenuated MPP^+^-induced oxidative damage, reduction of dopamine uptake, loss of TH-immunopositive neurons and degeneration of TH-immunopositive neurites, and 200 µg/mL of GPS had the maximum protective effect.	[91]
6-OHDA-induced rat model of PD	GPS and GP-EX	GPS (25 and 50 mg/kg) or GP-EX (50 mg/kg) for 22 days, p.o.	Attenuated the development of L-DOPA-induced dyskinesia without compromising the anti-parkinsonian effects of L-DOPA by modulating the biomarker activities of ΔFosB expression and ERK1/2 phosphorylation.	[92]
MPTP-induced rat model of PD	GPS	50 mg/kg for 21 days, p.o.	Improved the levels of TH-immunopositive cells and dopamine in the substantia nigra and striatum, suppressed NMDA receptor expression, and increased the phosphorylation of ERK1/2 and CREB in the |hippocampus, ameliorated deficits in habit learning and spatial memory by increasing the activation of the dopaminergic neuronal system and modulating NMDA receptor-mediated signaling systems.	[93]
MPTP-induced rat model of PD	GPS and GP-EX	GPS (50 mg/kg), GP-EX (50 mg/kg) for 21 days, p.o.	Improved the symptom of anxiety disorders by modulating the brain levels of dopamine and serotonin, increased the number of TH-immunopositive neuronal cells, and increased the density of dopamine neurons in the substantia nigra.	[94]
6-OHDA-induced rat model of PD	GP-EX	10 mg/kg and 30 mg/kg for 28 days, p.o.	Ameliorated the reduction of TH-immunopositive neurons, recovered the levels of dopamine, 3,4-dihydroxyphenylacetic acid, homovanillic acid, and norepinephrine in the striatum.	[95]

**Table 4 molecules-26-06249-t004:** Effects on ischemia–reperfusion(I/R) injury.

Active Composition	Models	Dosage	Activity/Mechanism	Ref.
GPS	Oxygen-glucose deprivation– reoxygenation (OGD/R) H9c2 cell model	5, 10, 20 µM, pretreatment prior to ischemia	Enhanced the cell viability, inhibited the translocation of NF-κB subunit p65 into nuclei, inhibited NF-κB activation by suppressing the MAPK signaling transduction pathway.	[97]
	Myocardial I/R rat model	50, 100, 200 mg/kg, pretreatment prior to ischemia, p.o.	Alleviated the impairments on the cardiac function and structure, suppressed the activation of NF-κB via inhibition of phosphorylation of inhibitor of NF-κB α (IκBα), ERK, JNK, and p38.	
GPS	OGD/R H9c2 cell model	5, 10, 20 µM, pretreatment prior to ischemia	Protected cell viability, reduced ROS production and oxidative stress, and preserved mitochondrial function.	[98]
	Myocardial I/R rat model	50, 100, 200 mg/kg, pretreatment prior to ischemia, p.o.	Attenuated infarct size, alleviated I/R-induced pathological changes in the myocardium, and preserved left ventricular function; reduced oxidative stress, preserved mitochondrial function in the cardiomyocytes, maintained mitochondrial membrane integrity, and inhibited the release of cytochrome c from the mitochondria to the cytosol.	
Gypenoside A	OGD/R H9c2 cell model	10, 20 µM, pretreatment prior to ischemia	Increases cell viability and inhibits apoptosis, exerted its protective effect on H9c2 cells by inhibiting the expression of miR-143-3p via the activation of AMPK.	[99]
GPS	OGD/R H9c2 cell model	10, 20 µM, pretreatment prior to ischemia	Exerted protective effect by inhibiting the ER stress-induced apoptosis via inhibition of CHOP pathway and activation of PI3K/Akt pathway.	[100]
	Myocardial I/R rat model	50, 100, 200 mg/kg, pretreatment prior to ischemia, p.o.		
Gypenoside	Renal I/R rat model	50 mg/kg, pretreatment prior to ischemia, i.v.	Inhibited I/R-induced up-regulation of serum creatinine and blood urea nitrogen, inhibited the production of pro-inflammatory cytokines, inhibited cell apoptosis by activating ERK signaling.	[101]
Gypenoside	Hepatic I/R rat model	50 mg/kg, pretreatment prior to ischemia, p.o.	Attenuated the increase in activity of serum aminotransferases in the hepatic tissue, attenuated the increase in hepatic lipid peroxidation and GSH content in hepatic tissue, up-regulated the protein expression of HO-1, attenuated I/R-induced hepatic cell apoptosis by modulating key apoptosis-related proteins.	[102]

**Table 5 molecules-26-06249-t005:** Hepatoprotective effects of GP.

Models	Medicinal Composition	Dosage	Activity/Mechanism	Ref.
CCl_4_-induced liver injury in rats	GPP	40 and 80 g/kg for 30 days, i.g.	Decreased the level of serum aspartate aminotransferase (AST) and alanine aminotransferase (ALT), down-regulated the iNOS mRNA expression in hepatic tissue, elevated the level of Bcl-2/Bax in hepatic tissue and alleviated liver injury.	[103]
CCl_4_-induced chronic liver injury in rats	Gypenoside	100 mg/5 mL/kg for 8 weeks, 4 times per week, p.o.	Reduced the activities of serum glutamic transaminase and serum glutamic pyruvate transaminase, elevated albumin/globulin ratio, prevented CCl_4_-induced liver damage, reduced the collagen content, suppressed the onset and enhanced the recovery of liver fibrosis induced by CCl_4_.	[104]
CCl_4_-induced acute liver injury in rats	GPS	100 mg/kg, i.v.	Decreased the level of serum ALT, ameliorated liver injury.	[105]
CCl_4_-induced liver fibrosis in rats	GPS	200 mg/kg for three weeks, i.g.	Ameliorated CCl_4_-induced liver fibrosis via the inhibition of TGF-β signaling, consequently inhibited the differentiation of hepatic progenitor cells into myofibroblasts.	[106]
Exhaustive exercise-induced hepatocyte apoptosis in rats	GPP	100 and 300 mg/kg, i.g.	Increased the expression of Bcl-2 and decreased the expression of Bax, increased SOD, GSH-Px, and CAT activities and decreased MDA contents in liver, exerted protective effect on oxidative stress and apoptosis induced by exhaustive exercise.	[107]
Alcohol-induced liver injury in rats	GPS	50, 100, and 200 mg/kg for 7 days, i.g.	Decreased the level of MDA and increased the activity of SOD, CAT, and glutathione, improved the level of the antioxidant defense system by activating the Nrf2 signaling pathway, reduced the level of inflammatory cytokines TNF-α and IL-6 by inhibiting the nuclear transposition of NF-kB.	[108]
High choline-induced liver injury in rats	GPS	200, 400, and 800 mg/kg for 8 weeks, i.g.	Elevated T-SOD and GSH-Px activity, decreased the MDA concentration, alleviated the high choline-induced oxidative stress injury, lowered the levels of AST and ALT.	[109]
High-fat and cholesterol diet and alcohol-induced fatty liver disease in rats	GPS	60, 30, and 15 mg/kg for 10 weeks, i.g.	Decreased the levels of serum TG, TC, LDL-C, free fatty acid, increased HDL-C level, increased SOD activity, reduced MDA level and ALT and AST activities, prevented liver fatty degeneration through up-regulation of PPAR-α expression in the liver to inhibit lipid peroxidation and hepatocyte apoptosis, ameliorated hepatic steatosis and damage, and improved hepatic function.	[110]
Methionine choline deficient diet-induced NAFLD in rats	UL4-rich GPE	50, 100, and 200 mg/kg for 8 days, i.g.	Reduced hepatic fat accumulation, hepatocellular injury, inflammation, and fibrosis, increased the expression of SIRT6 and phase 2 anti-oxidant enzymes, and then attenuated oxidative stress in the liver.	[111]

**Table 6 molecules-26-06249-t006:** Hypoglycemic effect of GP.

Models	Medicinal Composition	Dosage	Activity/Mechanism	Ref.
Alloxan-induced hyperglycemia in rats	Gypenoside	150, 250, 350 mg/kg for 4 weeks, i.g.	Exerted hypoglycemic effect by improving the synthesis of liver glycogen and protected the kidney.	[116]
L6 myotube cells	Damulin A	37.5, 75, 150 µM	Stimulated glucose uptake by stimulating the translocation of GluT4 to the cell membrane via AMPK activation.	[117]
	Damulin B	1.2, 6, 12 µM		
Alloxan-induced type 1 diabetes in rats	GPP	200 mg/kg for 5 d, i.g.	Lowered the fasting blood sugar and glucose tolerance, which might be related to the inhibitory effect on α-amylase.	[118]
High-sugar, high-fat diet combined with a small dose of streptozotocin-induced type 2 diabetes in rats	GPP	50, 100, and 200 mg/kg for 2 months, i.g.	Lowered the fasting blood sugar and glucose tolerance, which might be related to the increased level of serum insulin.	[119]
Alloxan-induced hyperglycemia in rats	GP-EX	1, 2, 4 g/kg for 10 days, i.g.	Lowered the level of blood glucose and had no effect on normal mice.	[120]
Glucose-induced hyperglycemia in rats				
Type 2 diabetic Goto-Kakizaki rats	GPE	1600 mg/kg/day for three weeks, p.o.	Reduced plasma glucose levels, suppressed hepatic glucose output levels significantly, and improved the hepatic insulin sensitivity by suppressing gluconeogenesis.	[121]

**Table 7 molecules-26-06249-t007:** Other activities of GP.

Other Bioactivities	Component/Dosage	Activity/Mechanism	Ref.
Anti-aging effect	GPS (8 g/kg) for 30 days, i.g.	Increased the activity of SOD, GSH-Px, and CAT in the skin, reduced the MDA content.	[133]
	GP (100 mg/kg) for 40 days, i.g.	Increased the activities of antioxidant enzyme SOD and GSH-Px, decreased the contents of NOS and NO in hypothalamus, reduced the strong excitatory toxicity of the two on neurons and delayed aging.	[134]
	GPP (50, 100, and 200 mg/kg) for 42 days, i.p.	Increased the activity of antioxidant enzymes (SOD, T-AOC, GSH-Px, CAT) in serum, brain and liver, increased the hydroxyproline content in the skin and reduced the damage caused by peroxidation products (MDA), thus delaying aging.	[135]
Anxiolytic effect	GPS (100, 200 mg/kg); GP-WX (50 mg/kg) for 10 days, p.o.	Recovered the number of open arm entries and the time spent on open arms, reduced the number of marbles buried, increased the spontaneous locomotor activities, increased the levels of dopamine and serotonin in the brain, reduced the serum levels of corticosterone, reduced c-Fos expression.	[136]
	GP-EX (400 mg) for 8 weeks, p.o.	Reduced “anxiety proneness” as shown by a decrease in the score of T-STAI and the tendency for a decrease in the total score of STAI.	[137]
Antidepressant effect	GPS (25, 50, and 100 mg/kg) for 4 weeks, p.o.	Increased the sucrose preference, reduced the immobility time, increased the hippocampal BDNF expression and neuronal proliferation, and exhibited antidepressant-like effects in mice, which might be mediated by activation of the BDNF-ERK/Akt signaling pathway in the hippocampus.	[138]
Immunomodulatory effect	GPP (50, 150, and 250 mg/kg) for 15 days, p.o.	Increased the spleen and thymus indices, activated the macrophages and NK cells, elevated CD4+ T lymphocyte counts as well as the CD4+/CD8+ ratio in a dose-dependent manner, and increased IL-2 level in the sera and spleen.	[139]
Anti-gastric ulcer effect	GP butanol fraction (200 and 400 mg/kg), p.o.	Preserved the gastric mucus synthesis and secretion but could not significantly decrease gastric volume and increase gastric pH and acidity.	[140]
Anti-fatigue effect	GPP (100, 200, 400 mg/kg) for 30 days, p.o.	Extended the exhaustive swimming time of the rats, elevated the exercise tolerance, lowered the levels of blood lactic acid and urea nitrogen, increased the hemoglobin, liver glycogen, and muscle glycogen concentrations, and postponed the appearance of fatigue.	[141]
	GPE (20 µg/mL) or GL (0.36 µg/mL)	Stimulated lactate metabolism, promoted the differentiation of myoblasts into myotubes, enhanced exercise endurance capacity.	[142]
Anti-platelet aggregation activity	GPS (10, 20, 40, and 80 mg/kg) for 4 days, i.p.	Inhibited ADP and collagen-induced platelet aggregation, inhibited thrombosis.	[143]
Sedative and hypnotic effect	GPS (50, 100 and 200 mg/kg) for 7 days, i.g.	Prolonged the sleep time, increased the contents of 5-hydroxytryptamine and IL-1β, decreased the contents of dopamine and noradrenalin, and showed no influence on glutamate and gamma-aminobutyric acid contents in the hippocampus.	[144]
Antimicrobial activity	GPE (100, 1000, 5000, and 10,000 ppm)	Showed antimicrobial activity against fungi producing aflatoxin and fumonisin and bacteria causing diarrheal disease.	[145]
Antiviral activity	GPP (150 mg/kg) for 3 days, p.o.	Protected MARC-145 cells from invasion by preventing PRRSV from absorbing marC-145 cells and exerted the best anti-infection effect when the concentration of GPP was 62.5 µg/mL.	[146]
	GP-EX and GP-WX	Inhibited the H5N1 virus replication in the MDCK cells.	[147]
Effect on hyperuricemia	GPS (15 and 60 mg/kg) for 8 weeks, i.g.	Decreased serum uric acid levels, promoted renal excretion of uric acid, increased the kidney index, down-regulated URAT1 and GLUT9 expression and up-regulated OAT1 expression in the kidney, decreased the levels of xanthine oxidase, adenosime deaminase, and xanthine dehydrogenase expression.	[148]
Effect on osteoporosis	12.5, 25 and 50 µM for 4 days	Inhibited osteoclast formation, inhibited osteoclastogenesis-related markers expression, such as MMP-9, c-Src, NFATc1, and cathepsin K, inhibited RANKL-induced NF-κB and MAPK activation and Akt phosphorylation.	[149]

STAI, State-Trait Anxiety Inventory; T-STAI, Trait Anxiety Scale of the STAI; ADP, Adenosine diphosphate; PRRSV, Porcine reproductive and respiratory syndrome virus; H5N1, Highly pathogenic avian influenza virus; T-AOC, Total antioxidant capacity; BDNF, Brain-derived neurotrophic factor; MDCK, Madin-Darby canine kidney; URAT1, Urate transporter 1; OAT1, Organic anion transporter 1; GLUT9, Glucose transporter 9; RANKL, Receptor activator of nuclear factor-kappa B Ligand; NFAT, Nuclear factor of activated T cells.

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
