# Peer review of "Progress in the Medicinal Value, Bioactive Compounds, and Pharmacological Activities of Gynostemma pentaphyllum"

_molecules, 2021, doi:10.3390/molecules26206249_

Round 1

Reviewer 1 Report

In the submitted manuscript, the authors studied the chemical constitutes and biological effects of Gynostemma pentaphyllum. The review is quite interesting, but the author needs to improve some issues in the manuscript.

  1. In the manuscript, the authors discussed various pharmacological effects of GP, especially details about Cancer and the Anti-atherogenic effect; moreover, there also mentioned weight loss efficacy in the abstract. The authors did not discuss obesity properly. I would like to suggest authors add the effect of GP on key regulators (UCP1, PRDM16, PGC-1α, and PPARγ) of obesity to improve the manuscript.
  2. The authors can modify figure 4 as like figure 6.

In the submitted manuscript, the authors studied the chemical constitutes and biological effects of Gynostemma pentaphyllum. The review is quite interesting, but the author needs to improve some issues in the manuscript.

  1. In the manuscript, the authors discussed various pharmacological effects of GP, especially details about Cancer and the Anti-atherogenic effect; moreover, there also mentioned weight loss efficacy in the abstract. The authors did not discuss obesity properly. I would like to suggest authors add the effect of GP on key regulators (UCP1, PRDM16, PGC-1α, and PPARγ) of obesity to improve the manuscript.
  2. The authors can modify figure 4 as like figure 6.

Author Response

In the submitted manuscript, the authors studied the chemical constitutes and biological effects of Gynostemma pentaphyllum. The review is quite interesting, but the author needs to improve some issues in the manuscript.

  1. In the manuscript, the authors discussed various pharmacological effects of GP, especially details about Cancer and the Anti-atherogenic effect; moreover, there also mentioned weight loss efficacy in the abstract. The authors did not discuss obesity properly. I would like to suggest authors add the effect of GP on key regulators (UCP1, PRDM16, PGC-1α, and PPARγ) of obesity to improve the manuscript.

Response:  Thank you for your professional and insightful comments. We complemented the anti-obesity effect of GP in our revised manuscript according to your suggestion.

  1. The authors can modify figure 4 as like figure 6.

Response:  We modified Figure 4 as like Figure 6 in our revised manuscript according to your suggestion.

Reviewer 2 Report

The article is well written however the following points may improve it further

  • Gynostemma pentaphyllum (Thumb.) Makino, check and confirm whether it is Thumb. Or Thumb.
  • The authors shall provide the methodology used for the data retrieval
  • The names of Gypenosides are not given in the manuscripts
  • Dementia is the term applied to a group of symptoms that negatively impact memory, but Alzheimer's is a progressive disease of the brain that slowly causes impairment in memory and cognitive function. Why not to mention it under neuroprotective effects not antidementia effect
  • Some activities are not reported like inhibition of uric acid formation (Pang et al., 2017) and osteoporosis (Han et al., 2018)
  • The authors have missed to mention IC50s determined or the weaknesses or strengths of studies as there existed some studies on cytotoxicity of isolated saponins with no reported IC50 values

Shi, L., Cao, J.-Q., Shi, S.-M., Zhao, Y.-Q., 2011. Triterpenoid saponins from Gynostemma pentaphyllum. J. Asian Nat. Prod. Res. 13(2), 168-177

Shi, L., Meng, X.-J., Cao, J.-Q., Zhao, Y.-Q., 2012b. A new triterpenoid saponin from Gynostemma pentaphyllum. Nat. Prod. Res. 26(15), 1419-1422.

 Yin, F., Zhang, Y.-N., Yang, Z.-Y., Hu, L.-H., 2006. Nine new dammarane saponins from Gynostemma pentaphyllum. Chem. Biodivers. 3(7), 771-782.

  • or no positive control

Shi, L., Tan, D.-H., Yan, T.-C., Jiang, D.-H., Hou, M.-X., 2018. Cytotoxic triterpenes from the acid hydrolyzate of Gynostemma pentaphyllum saponins. J. Asian Nat. Prod. Res. 20(2), 182-187.

  • Why the authors have not given the doses of reported studies e.g. Antiparkinsonian studies:   pentaphyllum extract (50 mg/kg in 21 days, p.o.) Kim et al. (2017) and gypenosides (50 mg/kg in 21 days, p.o.) Zhao et al. (2017) and for other may also be provided
  • These articles may be added

Gynostemma pentaphyllum extract and Gypenoside L enhance skeletal muscle differentiation and mitochondrial metabolism by activating the PGC-1α

https://www.e-nrp.org/DOIx.php?id=10.4162/nrp.2021.15.e45

Characterization and Identification of the Chemical Constituents of Gynostemma pentaphyllum Using High Performance Liquid Chromatography – Electrospray

Author Response

The article is well written however the following points may improve it further

  • Gynostemma pentaphyllum (Thumb.) Makino, check and confirm whether it is Thumb. Or Thunb.

Response:  Thank you for your professional and careful comments. The corrected name of GP was Gynostemma pentaphyllum (Thunb.) Makino, we corrected it in our revised manuscript.

  • The authors shall provide the methodology used for the data retrieval

Response:  We complemented the methodology used for the data retrieval in our revised manuscript.

  • The names of Gypenosides are not given in the manuscripts

Response:  We complemented the names of gypenosides in the revised manuscript.

  • Dementia is the term applied to a group of symptoms that negatively impact memory, but Alzheimer's is a progressive disease of the brain that slowly causes impairment in memory and cognitive function. Why not to mention it under neuroprotective effects not antidementia effect

Response:  We now moved “effect on Alzheimer’s disease” under the section of “Neuroprotective effect”.

  • Some activities are not reported like inhibition of uric acid formation (Pang et al., 2017) and osteoporosis (Han et al., 2018)

Response:  We complemented the corresponding activities in the section of “Others” in our revised manuscript.

  • The authors have missed to mention IC50s determined or the weaknesses or strengths of studies as there existed some studies on cytotoxicity of isolated saponins with no reported IC50 values

Shi, L., Cao, J.-Q., Shi, S.-M., Zhao, Y.-Q., 2011. Triterpenoid saponins from Gynostemma pentaphyllum. J. Asian Nat. Prod. Res. 13(2), 168-177

Shi, L., Meng, X.-J., Cao, J.-Q., Zhao, Y.-Q., 2012b. A new triterpenoid saponin from Gynostemma pentaphyllum. Nat. Prod. Res. 26(15), 1419-1422.

 Yin, F., Zhang, Y.-N., Yang, Z.-Y., Hu, L.-H., 2006. Nine new dammarane saponins from Gynostemma pentaphyllum. Chem. Biodivers. 3(7), 771-782.

  • or no positive control

Shi, L., Tan, D.-H., Yan, T.-C., Jiang, D.-H., Hou, M.-X., 2018. Cytotoxic triterpenes from the acid hydrolyzate of Gynostemma pentaphyllum saponins. J. Asian Nat. Prod. Res. 20(2), 182-187.

Response:  We complemented the corresponding IC50 value and weakness of studies which did not mentioned IC50 values or positive control in the section of “Relationship between anticancer activity and structure of GPS” in our revised manuscript.

  • Why the authors have not given the doses of reported studies e.g. Antiparkinsonian studies:   pentaphyllum extract (50 mg/kg in 21 days, o.) Kim et al. (2017) and gypenosides (50 mg/kg in 21 days, p.o.) Zhao et al. (2017) and for other may also be provided

Response:  We complemented the doses of several reported studies in our revised manuscript.

  • These articles may be added

Gynostemma pentaphyllum extract and Gypenoside L enhance skeletal muscle differentiation and mitochondrial metabolism by activating the PGC-1α

https://www.e-nrp.org/DOIx.php?id=10.4162/nrp.2021.15.e45

Characterization and Identification of the Chemical Constituents of Gynostemma pentaphyllum Using High Performance Liquid Chromatography – Electrospray

Response:  We added these two references in the revised manuscript.